# Genomic Instability and Replicative Stress in Multiple Myeloma: The Final Curtain?

**DOI:** 10.3390/cancers14010025

**Published:** 2021-12-22

**Authors:** Oronza A. Botrugno, Giovanni Tonon

**Affiliations:** 1Functional Genomics of Cancer Unit, Experimental Oncology Division, IRCCS San Raffaele Scientific Institute, 20132 Milan, Italy; 2Center for Omics Sciences, IRCCS San Raffaele Scientific Institute, 20132 Milan, Italy

**Keywords:** Multiple Myeloma, genomic instability, replicative stress, ATR, synthetic lethality

## Abstract

**Simple Summary:**

Genomic instability is recognized as a driving force in most cancers as well as in the haematological cancer multiple myeloma and remains among the leading cause of drug resistance. Several evidences suggest that replicative stress exerts a fundamental role in fuelling genomic instability. Notably, cancer cells rely on a single protein, ATR, to cope with the ensuing DNA damage. In this perspective, we provide an overview depicting how replicative stress represents an Achilles heel for multiple myeloma, which could be therapeutically exploited either alone or in combinatorial regimens to preferentially ablate tumor cells.

**Abstract:**

Multiple Myeloma (MM) is a genetically complex and heterogeneous hematological cancer that remains incurable despite the introduction of novel therapies in the clinic. Sadly, despite efforts spanning several decades, genomic analysis has failed to identify shared genetic aberrations that could be targeted in this disease. Seeking alternative strategies, various efforts have attempted to target and exploit non-oncogene addictions of MM cells, including, for example, proteasome inhibitors. The surprising finding that MM cells present rampant genomic instability has ignited concerted efforts to understand its origin and exploit it for therapeutic purposes. A credible hypothesis, supported by several lines of evidence, suggests that at the root of this phenotype there is intense replicative stress. Here, we review the current understanding of the role of replicative stress in eliciting genomic instability in MM and how MM cells rely on a single protein, Ataxia Telangiectasia-mutated and Rad3-related protein, ATR, to control and survive the ensuing, potentially fatal DNA damage. From this perspective, replicative stress *per se* represents not only an opportunity for MM cells to increase their evolutionary pool by increasing their genomic heterogeneity, but also a vulnerability that could be leveraged for therapeutic purposes to selectively target MM tumor cells.

## 1. Introduction

Multiple Myeloma (MM) is a hematological malignancy characterized by the accumulation of malignant plasma cells in the bone marrow and, during the progression of the disease, in the peripheral blood and other extramedullary sites. MM is associated with the production of immunoglobulin monoclonal proteins (M-protein), which are used as a reliable biomarker for the number of tumor cells in the body and are measured for clinical purposes in the serum and/or urine [1]. The accumulation of cancer cells is responsible for some of the clinical manifestations of the disease, including anemia, bone marrow failure, and bone destruction, which often results in hypercalcemia, that aggravates renal insufficiency, almost always occurring with the increase in the serum monoclonal (M) protein [2]. MM presents a median age at diagnosis of 69 years. In the USA, MM is the second most frequent hematological cancer after lymphomas, accounting for 18% of hematological malignancies, 1.8% of all new cancer cases, and 2% of all cancer deaths (Stat Fact Sheets SEER. Myeloma. http://seer.cancer.gov/statfacts/html/mulmy.html. accessed on 12 November 2021; [3]). MM is always preceded by a premalignant state, monoclonal gammopathy of undetermined significance (MGUS), which, in a small subset of patients, evolves towards an indolent form, smoldering myeloma (SMM), and finally to MM [4].

Molecular profiling technologies, including fluorescence in situ hybridization (FISH), microarray gene expression profiling (GEP), array comparative genomic hybridization (CGH), single-nucleotide polymorphism (SNP) arrays, and, recently, next generation sequencing (NGS), have brought a fairly comprehensive view on the genomic landscape of MM [5]. This knowledge has led to the identification of several molecular drivers triggering MM initiation and progression but has, disappointingly, failed to provide effective therapeutic targets or even biomarkers that could be used to dictate treatment strategies [5]. Conversely, other approaches have proven quite effective in delaying the progression of the disease, including treatments addressing MM non-oncogenic addictions [6], the interactions of MM cells with the microenvironment, and the presence of surface markers on MM plasma cells. Accordingly, the introduction of proteasome inhibitors, which hamper protein clearance inside plasma cells already overwhelmed by the production of immunoglobulins [7,8]; of lenalidomide-based treatments, which target for degradation essential factors for MM cell survival, including the Ikaros family zinc finger proteins 1 and 3 (IKZF1 and IKZF3), but also interfere with angiogenesis [9,10]; and most recently, of anti-CD-38 monoclonal antibodies [11,12,13,14], which have led to an important improvement in the survival of MM patients. However, despite this impressive evolution of treatment regimens, MM remains an incurable disease [15], warranting new therapeutic strategies. Notably, patients presenting with the unfavorable disease have benefited the least from the introduction of new drugs, thus representing still a highly vulnerable population [15]. 

Genomic instability has been recognized as a driving force in epithelial cancers [16], which, on one side, is an asset for cancer cells, broadening their evolutionary pool, but on the other is a liability. Conversely, blood cancers present a much more stable genome when compared with epithelial cancers [17], suggesting that genomic instability, and the ability to exploit it, may not represent a feature of hematological tumors. Yet, recent evidence accrued, especially in MM, would suggest otherwise. In this review, we will attempt to provide the groundwork for supporting the notion that blood cancers, in particular, MM, are endowed with rampant genomic instability rooted in replicative stress, which could be exploited for therapeutic reasons.

## 2. Genomic Instability in Cancer: We Know What It Is, We Do Not Know Where It Is Coming from

The existence of genomic instability in cancer has been debated for decades. It was unclear whether the multitude of genomic lesions present in cancer genomes resulted from a sudden, short-lived “big bang” [18,19,20,21], disrupting genome integrity, or was instead the result of a steadier, incremental accumulation of genetic lesions, reflecting an inherent, and cancer-specific, enhanced tendency to acquire genomic alterations over time [16]. It is increasingly apparent that both mechanisms could occur in cancer cells, with the latter labeled as genomic instability [16], which is now included among the enabling characteristics of cancer [22]. 

Why did it take so long to solve this debate? One possible explanation lies with the assay, as even measuring genomic instability is challenging. A common misconception equates genomic instability with the number of genetic lesions present at a defined point and time in a cell, hence, a state and not a rate (as aptly noted by Bert Vogelstein and collaborators several years ago [23]). As such, the assays currently used to define genomic alterations are descriptive of the cell (and the genome) status at a certain time and do not register ongoing changes. One potential assay, which may more precisely register instability, albeit not perfectly, is the measure of phosphorylated H2AX (γH2AX), which may quantify at least a specific subtype of instability. When γH2AX foci are more intense and numerous in cancer cells when compared with the corresponding normal tissues [24,25,26], this phenomenon may record an ongoing disruption of DNA. It should be noted, however, that this assay measures just a subset of all the potential phenotypes associated with genomic instability. For example, it does not measure the increased mutational load emerging from mismatch repair or chromosomal instability elicited by mitotic disruption.

As for the causes of genomic instability, in hereditary tumors, germline mutations in the so-called caretaker genes, such as BRCA1 and BRCA2, have been reported and linked to genomic instability [22].

However, most cancers are sporadic. In this setting, what is the cause of genomic instability? Based on observations gathered at the transition between benign and malignant cancers [27,28,29], a hypothesis has been put forward which blames the very genes engaged early in cancer development as the major culprits of genomic instability in sporadic cancers. The theory goes that oncogenes activation (or inactivation of tumor suppressors which control the cell cycle progression) deregulates DNA replication [29], eliciting replicative stress, which, in turn, causes genomic instability.

## 3. Replicative Stress as a Key Driver of Genomic Instability in Cancer

DNA replication is a tightly regulated process, as even a modest disruption may lead to fatal consequences. As such, cells devote a considerable amount of energy and an exquisitely fine-tuned molecular machine and several checkpoints to ensure its accuracy [30]. 

What, then, is replicative stress? The term broadly defines impediments to DNA replication, which result in the stalling and eventual collapse of the replication forks, causing DNA damage [31]. The causes of replication fork stalling are several, ranging from inappropriate origin firing to the presence of unresolved DNA secondary structures, even exhaustion of the nucleotide pools available for DNA synthesis or the presence of DNA–RNA hybrid intermediates [32] (Figure 1). An intriguing mechanism leading to fork stalling relates to the presence of highly expressed genes, which challenge replisome progression, triggering replication–transcription machinery collisions [32,33,34]. More specifically, DNA replication and transcription machinery during the S phase share the same DNA template, and defects in their coordination generate transcription–replication conflicts (TRCs). These conflicts occur when the replication forks encounter the RNA polymerase, thus stalling. This favors the transient formation of R-loops, nucleic acid structures composed of a DNA–RNA hybrid and the displaced non-template single-stranded DNA (ssDNA) [35,36], which is particularly prone to breaks due to the high accessibility of the ssDNA to metabolites, reactive oxygen species (ROS), and nucleases. 

The induction of DNA double-strand breaks (DSBs) following activation of many oncogenes is the result of their ability to lead to the pervasive formation of stalled forks which eventually collapse [31]. Several mechanisms can contribute to oncogene-induced DNA replicative stress. For example, activation of oncogenic RAS driving proliferation and cell growth increases the number of active replicons and leads to asymmetric fork progression [28]. It should be noted that RAS, like other oncogenes, might elicit the fork stalling through additional mechanisms. RAS, for example, also suppresses nucleotide metabolism by downregulating the ribonucleotide reductase subunit M2 (RRM2), a rate-limiting protein in dNTP synthesis, and causing dNTP pool depletion and premature termination of replication forks [37]. Moreover, RAS overexpression leads to increased global transcription activity to face the additional demands stemming from the increased proliferation [38], which is then linked to replicative stress due to R-loop accumulation and increased expression of the general transcription factor TBP [38].

The oncogene MYC, as well, induces replicative stress through various mechanisms. MYC overexpression causes the direct deregulation of DNA replication dynamics due to interaction with members of the DNA replication machinery [39,40,41,42]. Additionally, MYC is a transcription factor that controls the expression of a large fraction of cellular genes linked to cell cycle control. MYC may promote cell cycle progression and replicative stress through indirect mechanisms, by regulating the expression of genes involved in cellular proliferation [43] and DNA replication, including the majority of genes involved in the nucleotide biosynthesis pathway [44]. Finally, MYC also induces replicative stress through the generation of TRCs [44]. 

There are also additional mechanisms by which oncogenes may induce replicative stress. Aberrant oncogene expression results in the loss of redox homeostasis with the unbridled generation of ROS [45]. Accumulation of ROS leads to the formation of 8-oxoguanine, and the resulting oxidative DNA damage causes the replication fork to stall at lesions, ultimately promoting replicative stress in cancer cells [42,46,47]. 

Finally, the induction of DNA DSBs can also be observed when tumor suppressors are lost. The deregulated activity of pRB, p53, p16, and p14^ARF^ have been linked to replicative stress through various mechanisms that result in the promotion of G1-S transition [48].

## 4. Genomic Instability and Replicative Stress in Multiple Myeloma

So, how does this model also fit a slow-growing hematological cancer like multiple myeloma?

Very surprisingly, a few years ago, we and others discovered that MM cell lines, as well as patient cells, display exceedingly high levels of rampant DNA damage and DSBs, as assessed by the presence of γH2AX foci in the absence of exogenous stressors [49,50,51,52,53,54]. DNA damage is absent in normal plasma cells or B-cells but appears in MGUS samples and increases in MM patient samples and almost all reported MM cell lines [49,50,54]. This genomic instability was linked to replicative stress since MM cells with ongoing DNA damage demonstrated positivity to a panel of replicative stress markers, including the phosphorylated form of RPA32, a subunit of replication protein A (RPA) [50]. Accordingly, cell cycle regulatory genes and genes related to DNA replication were significantly altered in MM cells. Importantly, the cells of a subset of MM patients presented with a signature of chromosomal instability and increased expression of replicative genes; these patients presented a poor prognosis [50]. 

What could be the cause of this replicative stress? MYC has been long known as a crucial gene in MM progression [55,56,57]. Surprising recent observations [58,59,60,61], however, point to a role for MYC in the early phases of MM. Indeed, patients whose cancer cells presented with increased replicative stress showed enhanced expression of the oncogene MYC. The silencing of MYC in MM cells presenting DNA damage reduced the number of γH2AX foci, whereas MYC overexpression in the U266 MM cell line, endowed with low levels of DNA damage and no c-MYC genomic rearrangements nor MYC overexpression, triggered DSBs [50]. In all, these results suggest that MYC, a crucial gene in MM, fosters replicative stress in these cells.

It remains mysterious how a slow-growing, tumor-like MM could present such a high degree of replicative stress, despite the overexpression of the MYC oncogene. Likely, some of the mechanisms included in Figure 1 are not per se linked to enhanced proliferation and may elicit replicative stress, and thus genomic instability, in the absence of an apparent increased proliferation.

While a large array of oncogenes is genetically altered and/or overexpressed in MM, it is tempting to speculate that at least a few of them may be causative of replicative stress in this disease, including cyclins, present at early-occurring chromosomal translocations or overexpressed in the absence of clear genetic lesions, or the oncogenes MAF and MAFB [62]. Notwithstanding, the significant association between MYC expression and replicative stress, and the absence of replicative stress in a few MM cell lines without c-MYC overexpression, all point to a crucial role for this gene in triggering replicative stress in this disease.

## 5. ATR, the Last Hope

What is the physiological response to replicative stress? Several lines of evidence point to a single protein as the central node that coordinates the cellular response to replicative stress. This is ataxia telangiectasia-mutated and Rad3-related protein, ATR.

ATR belongs to the family of phosphoinositide 3-kinase (PI3K)-related kinases (PIKKs), which also includes ATM (ataxia telangiectasia mutated) and DNA-PK (DNA-dependent protein kinase) [63]. These proteins control the DNA damage response, eliciting and regulating DNA repair and the associated events [63]. In particular, DNA-PK and ATM are crucial for DSBs repair by two different pathways, non-homologous end joining (NHEJ) and homologous recombination (HR), respectively. ATR is instead recruited to an extended tract of ssDNA coated with RPA [64] and is specialized in addressing replicative stress.

ATM and DNA-PK genes are mutated [65,66] in various Mendelian genetic diseases, while germline mutations affecting ATR have been linked to only one hereditary disorder, Seckel syndrome [67,68]. The fact that mutations causing Seckel syndrome are hypomorphic, with only a partial loss of gene function [67], and that ATR knockout mice are embryonic lethal further implies that ATR is an essential gene [69,70].

ATM germline mutations increase cancer susceptibility, and ATM is frequently mutated in a broad range of sporadic cancers [71]. Genetic alterations of PRKDC, the gene encoding the catalytic subunit of DNA-PK, are common in several cancer types [72]. Again, somatic mutations in the ATR gene are exceedingly rare [73] and no cancer predisposition has been reported for patients with Seckel syndrome or in Seckel mice, even in a p53 mutant background [74]. In fact, several lines of evidence suggest that cancer cells require a functional ATR to withstand the intense replicative stress elicited by the cancer genes deregulation [75].

How does ATR work? ssDNA stretches are promptly coated by RPA, which prevents ssDNA breakage and provides a platform for the recruitment of accessory factors that, in turn, activates ATR through phosphorylation and its downstream effectors, such as Checkpoint kinase 1 (CHK1) [75]. Activated ATR coordinates the signaling cascade that triggers S and G2/M cell cycle checkpoints, stabilizes the replicative forks promoting forks repair through the engagement of the Fanconi anemia pathway, and reduces replicative stress [76]. Accordingly, tumors with high levels of replicative stress have been shown to be particularly vulnerable to the loss of ATR [77]. In mice, where ATR expression has been reduced by 90% (“hypomorphic”), the tumor growth elicited by the expression of oncogenes was reduced, with minimal impact on normal tissue homeostasis [78,79].

## 6. Harnessing Replicative Stress as a Cancer Vulnerability in MM

Given the widespread presence of replicative stress in cancer, several attempts have been made to therapeutically target it.

Adding up the replicative stress already elicited by activated oncogenes, exogenous genotoxic agents, including chemotherapeutic drugs, often trigger additional stress, introducing lesions in the template DNA, by several means (Figure 1). Nucleoside analogues exert their cytotoxic effects, competing with natural nucleotides for incorporation into DNA during replication. These fraudulent nucleotides, once incorporated into replicating DNA, give rise to a steric impediment for the extending replication forks, which results in the termination of DNA replication, fork stall, and accumulation of ssDNA. In addition, some nucleoside analogues, such as gemcitabine or 5-fluorouracil, that inhibit, respectively, the enzymes ribonucleotide reductase and thymidylate synthetase, key regulators of dNTP biosynthesis in mammals, reduce the nucleotide pools available for DNA synthesis [80,81]. Topoisomerase inhibitors also induce replicative stress [48]. Topoisomerases enzymes alter the topological state of DNA by catalyzing the relaxation of supercoiled DNA during cellular processes, such as replication and transcription, through the cleavage and then religation of one (type I) or both (type II) of the DNA strands [82]. Topoisomerase inhibitors typically associate with their target enzymes as they bind to DNA, which creates a barrier to ongoing replication forks [48]. Finally, alkylating agents and platinum-based drugs directly modify the DNA through the formation of intra- and inter-strand crosslinks between bases (ICLs), thus generating a physical barrier for the progression of the replication fork [83,84].

Despite their effectiveness, one major limitation of traditional anticancer drugs is represented by their poor specificity that is commonly associated with unbearable toxicity. For this reason, given the central role of ATR in mending the damages generated by replicative stress, considerable efforts have been devoted to the identification of specific ATR inhibitors. The development of small molecules targeting ATR faced technical hurdles, as the generation of appropriate biochemical kinase assays and cell-based screenings proved to be challenging [77]. Caffeine and Schisandrin B were the first compounds reported to inhibit ATR activity, albeit at very high doses and with limited selectivity [85,86]. The development of a cell-based platform for microscopically assessing and quantifying pan-nuclear γH2AX as a readout of ATR activity [87] allowed the identification of ETP-46464, the first potent ATR inhibitor with no action against the other PIKKs [88]. The development of a high throughput screening, based on a recombinant ATR kinase allowing the combination of a structure–activity relationship and homology modeling studies, led to the discovery of the first series of potent and selective ATR inhibitors [89], with one of them, VE-821, further progressing preclinical evaluation as a sensitizer of radiotherapy and chemotherapy treatments in several tumor models. The improvement of the pharmacological properties of VE-821 led to the development of the optimized analogue VE-822/VX-970/M6620/berzosertib with increased potency and selectivity against ATR and better pharmacokinetic profile and bioavailability, which has allowed the in vivo evaluation of ATR inhibition [77,90].

Following these initial studies, several structurally unrelated ATR inhibitors have been described and widely assayed in multiple preclinical models of cancers [91,92,93,94,95,96]. Recently, phase I/II cancer clinical trials have been launched to assess the role of ATR inhibitors, either as single agents or in combinations with chemotherapy, radiotherapy, and immunotherapy in patients with advanced and refractory solid tumors (clinicaltrials.gov and [97]), with the first results reported [98,99,100,101].

ATR activity was also found to be central for hematological cancers [78]. In this setting, it was reported that ATR activity is required to compensate for MYC-induced replicative stress. ATR downregulation or pharmacological inhibition prevented the repair of stalled replication forks, increasing DNA damage and triggering apoptosis [50,53]. Moreover, inhibition of ATR selectively targets the subset of MM tumors endowed with increased replicative stress and ongoing DNA damage [50]. The MYC oncogene again seems to exert a central role in replicative stress in MM. Notably, by combining ATR inhibition with ROS induction, using piperlongumine, apoptosis was greatly increased [50]. Along similar lines, in a more clinically relevant setting, the combination of ATR inhibition with VX-670/M6620 and the alkylating agent melphalan, one of the key drugs used in MM chemotherapy, was strongly synergistic, both in vitro and in vivo [53]. This synergy was evident even in cases resistant to the treatment with melphalan alone, suggesting a potential role for ATR inhibition in patients resistant to ICLs-inducing drugs, such as melphalan [53].

A handful of genes confer a predisposition towards the development of MM, including, for example, germline mutations in ARID1A [102], whose aberrant activity has been shown to lead to DNA replication stress associated with augmented R-loops formation and TRCs in human cells [103]. It remains to be tested whether tumors arising in this genetic background may benefit from ATR inhibition as well.

CHK1 is the main downstream target of ATR, thus providing an alternative option for targeting the ATR pathway [104]. CHK1 is crucial for a proficient replicative stress response. Activated CHK1 inhibits CDC25, which positively regulates cyclin-dependent-kinases (CDKs), the major drivers of cell cycle progression. The CHK1 inhibitor AZD7762 induces replicative stress and increases ATR activity [105]. In line with the results obtained in MM with ATR inhibitors in combination with melphalan, AZD7762 has also been shown to potentiate the antiproliferative effects of bendamustine, melphalan, and doxorubicin in p53-deficient MM cell lines [106]. CHK1 also activates the Wee1-like protein kinase (WEE1) [107], which restricts mitotic entry by reducing the activation of CDKs [108]. So, an alternative or additional approach to target CHK1 is to inhibit WEE1, forcing cancer cells with high replicative stress to enter prematurely into mitosis and undergo apoptosis [48]. Moreover, activation of CDKs activity following WEE1 kinase inhibition has been shown to stimulate DNA replication, leading to nucleotide shortage, a reduction in replication fork speed, fork stalling, and, finally, DNA DSBs [109]. The reduced cellular nucleotide levels upon WEE1 inhibition resulted from RRM2 degradation through CDK activation [110]. These observations have supported the rationale for targeting WEE1 in combination with replicative stress-inducing agents in preclinical models that have shown encouraging efficacy [97]. Preclinical studies suggest, indeed, that WEE inhibitors have single agent and chemo-sensitizer effects also in MM [111,112,113,114].

## 7. Unexpected Travel Companions

Additional potential approaches to modulate DNA replicative stress for MM treatment rely on the inhibition of the enzymes belonging to the poly ADP-ribose polymerase (PARP) family, in particular PARP1. Besides its role in activating DNA repair programs and modulating cell death, PARP1 regulates the rate of replication fork progression, favoring the fork slowdown upon treatments hampering DNA replication [115], and enhancing CHK1 binding and activation at stalled forks, thus aiding forks to restart after release from the block [115,116]. In the absence of PARP1, ssDNA breaks require BRCA-mediated HR repair to prevent their degeneration into DSBs. This explains, in part, the cellular lethality observed when PARP1 inhibition is combined with the inactivation in genes involved in the HR repair pathway, as in the case of the BRCA deficient tumors [117,118]. In these cancers, the use of PARP inhibitors has become the mainstay treatment, as the inhibition of PARP1 is not compensated by a functional HR response.

The introduction of PARP inhibitors in the clinic provides a glimpse of a broader concept, that is, the relevance of synthetic lethality in cancer treatment [119]. This concept, derived from the genetic field, posits that two genes are synthetic lethal when inactivating mutations affecting each of them alone is compatible with viability, but mutations affecting both are incompatible with life, be it cellular or organismal. Thus, targeting a gene that presents synthetic lethality with a cancer-relevant mutation would lead to the demise only of cancer cells, sparing normal cells.

The synthetic lethal interaction between PARP inhibition and BRCA mutations has provided a strong impetus to test several combinations of PARP inhibitors in clinical trials. Of the six small-molecule PARP inhibitors available in the clinic, four (olaparib, rucaparib, niraparib, and talazoparib) were granted FDA and/or EMA approval for the use in patients with gynecological and breast cancers harboring *BRCA1*/*2* mutations [120].

As for the *BRCA1/2* status, additional mechanisms related to “BRCAness” have been explored, besides germline *BRCA1*/*2* mutations [121]. In fact, it has been noted that sporadic tumors not harboring loss-of-function mutations in the *BRCA1*/*2* genes nonetheless phenocopy a deficiency in HR repair, as seen in these hereditary cancers, indeed a “BRCAness” status [122]. A subset of MM patients presents similar features. Due to the high levels of replicative stress-associated DNA damage and the resulting genomic instability, MM cells are highly dependent on the HR machinery. Even a moderate reduction in HR gene levels has a negative impact on MM cells, whereas high expression has been associated with chemotherapy resistance [123]. Accordingly, bortezomib induces a “BRCAness” state in MM cells, since it gives raises to the accumulation of polyubiquitylated proteins in the cytoplasm, depletion of nuclear-free ubiquitin, and, as a consequence, the abrogation of H2AX polyubiquitylation, an essential step for the recruitment of BRCA1 and RAD51 to the sites of DNA DSBs and the initiation of HR-mediated DNA repair. Thereby, treatment with bortezomib in MM cells results is synthetic lethal when combined with PARP inhibitors [124]. Along the same lines, Dinaciclib, a potent small-molecule inhibitor of CDKs, has been shown to impair HR repair efficiency in MM cells and sensitizes them to PARP inhibition [125].

Recently, a correlation between high MYC expression and sensitivity to PARP inhibitors has been reported in MM [126]. Mechanistically, MYC overexpression in MM cells hampered the NHEJ, exposing MM cells to a new liability towards PARP inhibition. 

These findings have prompted the translation of these novel therapeutic approaches also in MM, leading to preclinical investigations of the potential role of PARP inhibitors in MM, for example, veliparib in combination with bortezomib as mentioned above [124], and olaparib as monotherapy [126].

It is now accepted that some PARP inhibitors in addition to being DNA repair inhibitors by simply inhibiting the catalytic activity of PARP1, work as DNA damaging agents thanks to their ability to stabilize PARP–DNA complexes at the site of DNA damage [127,128]. The “trapping” of PARP1 on DNA results in a highly cytotoxic protein–DNA complex, which induces replicative stress and may prevent access of DNA repair proteins to the site of the damage. Consistently, combinations of ATR and CHK1 inhibitors with PARP inhibitors show synergistic cell killing effects in several tumors [129]. Further studies are warranted to demonstrate that the combinations are synergistic in preclinical models of MM.

Other genes altered in MM have been implicated in the control and modulation of replicative stress. For example, the RECQ-like family helicases present some functional overlap with PARP1, including roles in stabilization and repair of damaged DNA replication forks, HR, and DNA damage checkpoint signaling [130,131]. In particular, RECQ1 helicase is overexpressed in MM and favors the recovery of MM cells from replicative stress, which finally confers higher resilience against chemotherapy [132]. RECQ1 depletion increases spontaneous DNA damage, either ssDNA breaks or DSBs accompanied by an impaired progression of replication forks, leading to MM cell apoptosis. Moreover, RECQ1 depletion significantly sensitized MM cells to PARP inhibitors-induced apoptosis [132].

The HECT-type ubiquitin ligase HUWE1 alleviates replicative stress through its interaction with the replication factor PCNA and the mono-ubiquitination of H2AX [133]. A more recent paper has highlighted the direct role of HUWE1 in regulating the protein levels of CHK1, independently from ATR, through polyubiquitination [134]. In MM, HUWE1 presents somatic mutations [135] and has been implicated in the DNA repair response [136].

RAD51 recombinase, a central protein in HR, highly expressed in MM cell lines and bone marrow aspirates, was suggested to mediate disease progression and chemotolerance [137]. Besides its role in repair, RAD51 has been implicated in several steps in response to replicative stress, including replication fork protection, remodeling, and restart [138], which implies it is a promising target for manipulating replicative stress levels in cancer cells. Chemical inhibition of RAD51 by the small-molecule inhibitor B02 or modulation of RAD51 expression both led to marked inhibition of MM cell survival in the absence of exogenous DNA damage and sensitized tumor cells to the topoisomerase II inhibitor doxorubicin and the alkylating agent melphalan through DSB induction and the subsequent blocking of HR repair [51,139,140].

Finally, inhibition of the exonuclease activity of MRE11 by mirin showed promising results in killing only myeloma cells displaying a high level of replicative stress and endogenous DNA damage, evidenced by the high amount of γH2AX and RAD51 foci [51]. 

## 8. Perspectives and Conclusions

The high levels of genetic and clonal heterogeneity and the lack of a unifying driver event represent essential challenges in the management of MM and potential explanations for the failure of the therapies directed to a specific set of mutations or more genetic lesions [141]. A more effective strategy could be to target a pathway to which MM shows addiction, irrespective of the mutations present in every single clone, somehow in line with the non-oncogene addiction paradigm [6]. The surprising finding that MM cells present a vigorous ongoing DNA damage, fueled by intense replicative stress, on one side explains the remarkable adaptability of MM cells to pharmacological and environmental hostile agents, and on the other exposes these cells to liabilities. In particular, to repair DNA, MM cells end up relying on a single protein, ATR, which is rarely, if ever, inactivated in cancer. Interfering with ATR has become one of the most enticing avenues to hamper the ability of MM cells to overcome the entrenched replicative stress and precipitate ruinous, irreversible DNA damage. Importantly for the practitioner, compounds targeting ATR present limited toxicity, since healthy tissues do not require as badly the activity of ATR for their survival. As such, replicative stress may become a fundamental vulnerability that could be therapeutically exploited either alone or in combinatorial regimens, thus hopefully bringing us a step closer to the final curtain for this deadly disease.

## Figures and Tables

**Figure 1 cancers-14-00025-f001:**
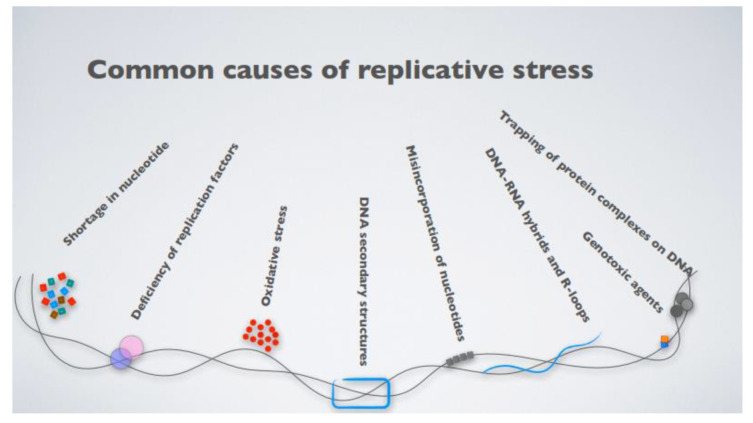
The most common causes contributing to cellular replicative stress.

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
