# Peer review of "Genomic Instability and Replicative Stress in Multiple Myeloma: The Final Curtain?"

_cancers, 2021, doi:10.3390/cancers14010025_

Round 1

Reviewer 1 Report

This is a very comprehensive review, elucidating new 

ly discovered pathogenetic pathways and explaining possible effects reverberating on therapy.

I have only two suggestions, useful in my opinion to increase the interest of clinical hematologists: if possible, a more detailed Summary; in any case, a more detailed Perspestives and Conclusion Section, with possibly useful insights for the Practitioner.

Author Response

This is a very comprehensive review, elucidating new ly discovered pathogenetic pathways and explaining possible effects reverberating on therapy.

I have only two suggestions, useful in my opinion to increase the interest of clinical hematologists: if possible, a more detailed Summary; in any case, a more detailed Perspestives and Conclusion Section, with possibly useful insights for the Practitioner.

We thank the reviewer for her/his positive assessment of our manuscript. We have now expanded the Summary and the Perspestives and Conclusion Section (in red), where we have now attempted to also introduce some details which we hope could provide interesting insights for our clinical colleagues.

Reviewer 2 Report

This is a very good review on genome instability elicited by replication stress in MM. The authors provide an excellent description of replication stress in MM. I have only few comments to some part.

Line 68-71---“Conversely, blood cancers present ~ MM would suggest otherwise.”

For this sentence, properly references are needed.

Line 132---DSBs

Please provide the full name of this abbreviation. (DNA double-strand breaks)

Line 199---ATR section

It is thought that many cancer cells have replication stress, and employ the ATR-CHK1 pathway to release it and to survival. I suggest authors to mention that ATR is amplified in many types of cancer, for example, by citing the cBioPortal database.

Line 376---8. Perspectives and Conclusions

This sentence should be in bold. Please leave one line between line 375 and line 376.

Reviewer 3 Report

In this paper, Oronza A. Botrugno  and Giovanni Tonon, made a review related with the current understanding of the role of replicative stress in MM genomic instability, and how this could be used for therapeutic purposes to selectively target the MM.

The subject of this article are interesting and actual. However, as it is a review paper, the authors could also explore better other related mechanisms as the role of E3 ligase HUWE1 and REC1 helicase. Further, the authors should clarify better the role of synthetic lethality and how this could be translated in clinical practice.

Furthermore, the conclusion needs to be improved including which combinatorial therapeutic regimens the authors proposed, and which of them they think to be more efficient.

Other comments:

  1. In introduction the authors said: “…while others (hypercalcemia and renal insufficiency) are due to the accumulation of the M protein”. This is not entirely correct, as hypercalcemia is mainly related with osteoclast activity. The authors should clarify this aspect.
  2. In line 57 the authors must actualized the regimens as now the lenalidomide-based regimes are more used than thalidomide-based regimens (see ESMO guidelines 2021).
  3. Item 2 need an improvement
  4. Figure 1 needs a title and a higher resolution as we can see what are written in each mechanism/cause.
  5. In item 7 the author could refer wich PARP inhibiots are already approved for other neoplasias and wich of them are under study for MM.

Author Response

In this paper, Oronza A. Botrugno  and Giovanni Tonon, made a review related with the current understanding of the role of replicative stress in MM genomic instability, and how this could be used for therapeutic purposes to selectively target the MM.

The subject of this article are interesting and actual. However, as it is a review paper, the authors could also explore better other related mechanisms as the role of E3 ligase HUWE1 and REC1 helicase. Further, the authors should clarify better the role of synthetic lethality and how this could be translated in clinical practice.

We thank the reviewer for pointing out the role of these two genes in replicative stress, we have now added a section describing the role of these genes in DNA damage, although we have some doubts whether the reviewer equates REC1 with the already included description of the RECQ1 gene.

Furthermore, the conclusion needs to be improved including which combinatorial therapeutic regimens the authors proposed, and which of them they think to be more efficient.

We have added a section on the available combinatorial regimens.

Other comments:

  1. In introduction the authors said: “…while others (hypercalcemia and renal insufficiency) are due to the accumulation of the M protein”. This is not entirely correct, as hypercalcemia is mainly related with osteoclast activity. The authors should clarify this aspect.

We have modified the text accordingly.

  1. In line 57 the authors must actualized the regimens as now the lenalidomide-based regimes are more used than thalidomide-based regimens (see ESMO guidelines 2021).

We have modified the text accordingly.

  1. Item 2 need an improvement

We have corrected a few inaccurate statements and clarified some sentences.

  1. Figure 1 needs a title and a higher resolution as we can see what are written in each mechanism/cause.

We have increased both the size font of the title, and of each text item, to make it more clear.

  1. In item 7 the author could refer wich PARP inhibiots are already approved for other neoplasias and wich of them are under study for MM.

We have reshuffled the text extensively, including also a discussion on synthetic lethality, as requested by this reviewer in the general section.